# Nitrogen Enriched Solid-State Cultivation for the Overproduction of Azaphilone Red Pigments by *Penicillium sclerotiorum* SNB-CN111

**DOI:** 10.3390/jof9020156

**Published:** 2023-01-24

**Authors:** Téo Hebra, Véronique Eparvier, David Touboul

**Affiliations:** 1Université Paris-Saclay, CNRS, Institut de Chimie des Substances Naturelles, UPR 2301, 91198 Gif-sur-Yvette, France; 2LCM, CNRS, Ecole Polytechnique, Institut Polytechnique de Paris, Route de Saclay, 91120 Palaiseau, France

**Keywords:** azaphilones, pigments, molecular networking, metabolomics, solid-state culture

## Abstract

Azaphilones are microbial specialized metabolites employed as yellow, orange, red or purple pigments. In particular, yellow azaphilones react spontaneously with functionalized nitrogen groups, leading to red azaphilones. In this study, a new two-step solid-state cultivation process to produce specific red azaphilones pigments was implemented, and their chemical diversity was explored based on liquid chromatography coupled to tandem mass spectrometry (LC-MS/MS) and a molecular network. This two-step procedure first implies a cellophane membrane allowing accumulating yellow and orange azaphilones from a *Penicillium sclerotiorum* SNB-CN111 strain, and second involves the incorporation of the desired functionalized nitrogen by shifting the culture medium. The potential of this solid-state cultivation method was finally demonstrated by overproducing an azaphilone with a propargylamine side chain, representing 16% of the metabolic crude extract mass.

## 1. Introduction

Azaphilones are polyketides produced by many genera of ascomycetes such as *Monascus*, *Aspergillus*, *Talaromyces* or *Penicillium* [1]. Depending on their subfamilies, azaphilones displays anti-inflammatory [2,3], antiviral [4], or antimicrobial [5] activities. They show strong yellow, orange, red or purple colors, leading to a wide usage in food industry as natural pigments [6]. As the need for new natural and nontoxic pigments increases with recent food regulations [7], azaphilones are proposed candidates as new biosynthetic pigments [8]. As azaphilones are only produced by microorganisms, the benefits of liquid and solid-state cultivation can be exploited such as easy handling, fast growth leading to large amount of material in a limited space and the fine tuning of the production process by controlling the chemical and physical conditions avoiding variations due to seasons (temperature, humidity, sunlight, etc.) compared to plants. 

*Monascus* is the most characterized genus in regard to industrial production of azaphilone pigments as it is involved in traditional food fermentation processes in Asia. Moreover, several industrial *Monascus* sp. strains, for which the biosynthesis pathways of azaphilone have been largely studied [9,10], are already commercially exploited in food industry [6]. However, the use of azaphilones produced by *Monascus* sp. has not been approved on the European market, mainly because of concerns about the co-production of citrinin, a mycotoxin linked to nephrotoxic, hepatotoxic and cytotoxic effects, during cultivation [8]. Some *Penicillium* sp. strains, such as *P. purpurogenum* or *P. sclerotiorum*, are known to produce azaphilones, such as PP-V or sclerotiorin, without citrinin co-production [11,12]. Together with the *Talaromyces* genus, they are proposed as alternatives to the *Monascus* genus for industrial sources of biosynthetic pigments [13]. Moreover, more than 14 amino analogs of sclerotiorin are reported in the literature, making them one of the most diverse subfamilies of azaphilone red pigments [14]. 

Many studies have focused on the overproduction of red azaphilone pigments by cultivation [15,16]. Among them, the influence of several parameters has been explored such as carbon sources [17], solid vs. liquid conditions [18], exposure to light [19], pH and the nitrogen sources [20]. In the two last years, processes of liquid submerged cultivation to obtain red azaphilone pigments with specific amino acids incorporation have been reported [21,22,23]. Nevertheless, in some particular cases, such as *Penicillium sclerotiorum* SNB-CN111 isolated from *Nasutitermes similis* termite aerial nests [24,25], the selected strain did not produce azaphilones in liquid potato dextrose media but in a solid one. For that purpose, we developed of a unique two-step nitrogen enriched solid-state cultivation method able to overproduce red azaphilone pigments, and their chemical diversity related to the nitrogen sources was characterized using liquid chromatography coupled to tandem mass spectrometry (LC-MS/MS) and molecular network by MetGem software [26,27,28] offering a new pipeline for large-scale red azaphilone production [25]. 

## 2. Materials and Methods

### 2.1. General Experimental Procedures

Optical rotations were measured at 20 °C in methanol using an Anton Paar MCP 300 polarimeter (Anton Paar, Graz, Austria) in a 100 mm long 350 μL cell. UV spectra for pure molecules were recorded at 20 °C in methanol using a PerkinElmer Lambda 5 spectrophotometer. NMR spectra were recorded on Bruker 500 MHz and 700 MHz spectrometers (Bruker, Rheinstetten, Germany). The chemical shifts (δ) are reported as ppm based on the solvent signal, and coupling constants (J) are in hertz. All solvents were HPLC grade, purchased from Sigma-Aldrich (Saint-Quentin-Fallavier, France).

### 2.2. Isolation and Identification of Termite Mutualistic Microorganisms

The strain was isolated from *Nasutitermes similis* termite aerial nest sampled in Piste de Saint-Elie (N 05 ° 01,838’ W 052 ° 44,606’) in French Guiana. The strain SNB-CN111 from the strain library collection at ICSN was identified as *Penicillium sclerotiorum*. The taxonomic marker analyses were externally performed by BACTUP Saint-Priest, France). The identification of the fungi was conducted by amplification of the ITS4 region of ribosomal. The sequences were aligned with DNA sequences from GenBank, NCBI (http://www.ncbi.nlm.nih.gov, accessed on 7 June 2021), using BLASTN 2.2.28. The sequence has been registered in the NCBI GenBank database (http://www.ncbi.nlm.nih.gov, accessed on 7 June 2021) under registry number KJ023726. 

### 2.3. Culture and Extraction of Microorganisms

#### 2.3.1. General Cultivation and Extraction Procedure

*Penicillium sclerotiorum* SNB-CN111 strain was cultivated on solid Czapek medium (CZK), potato dextrose (PD) or PD supplemented with yeast extract (YE) medium at 26 °C for 15 days, on a single Petri dish of 10 cm diameter (~80 cm^2^). All nitrogen sources (ammonium acetate (CAS 631-61-8, Sigma-Aldrich, Saint-Quentin-Fallavier, France), ethanolamine (Sigma-Aldrich, CAS 141-43-5) and propargylamine (Sigma-Aldrich, CAS 2450-71-7)) were prepared as 10× solution in milli-Q water, filtered on sterile condition on 0.22 µm membranes, and added to the CZK medium after autoclaving, once the temperature was 40 °C. The contents of the Petri dishes were transferred into a large container and macerated with ethyl acetate for 24 h. The organic solvent was collected by filtration, washed with water in a separating funnel and evaporated to dryness under reduced pressure. 

#### 2.3.2. Cellophane Membrane Preparation

Cellophane membranes were prepared as described by Fauchery et al. [29]. They were trimmed to the size of the Petri dish, put into boiling distilled water containing EDTA (1 g.L^−1^) for 20 min in order to permeate the membrane, rinsed four times in a big container of milliQ water and autoclaved. 

### 2.4. Isolation of Compound 63

The crude extracts of *Penicillium sclerotiorum* SNB-CN111 from CZK medium supplemented with propargylamine were pulled together (120 mg) then fractionated by reversed-phase flash chromatography (Grace Reveleris, Grace, Maryland, USA) using a 40 g C18 column and ultraviolet (UV) detectors. A linear gradient of H_2_O/formic acid (99.9/0.1) (A)—acetonitrile/formic acid (99.9/0.1) (B) (from 5% B to 100% B in 60 min, flow rate at 30 mL.min^−1^) was performed to generate 12 fractions labeled F1 to F12. Fraction of interest F1 (30 mg) was submitted to preparative HPLC. Further fractionation of F1 (H_2_O/formic acid (99.9/0.1) (A)—acetonitrile/formic acid (99.9/0.1) (B) isocratic, 65% B for 15 min led to the isolation of the compound **63** (19 mg, t_R_ = 11.1 min).

Isochromophilone XVI (**63**): Red amorphous oil; [α]^20^_D_ 600 (*c* 0.1 g.L^−1^, MeOH), UV (MeOH) λmax (ε), 367 nm (21,300 L.mol^−1^.cm^−1^), 480 nm (3800, L.mol^−1^.cm^−1^); ^1^H NMR (500 MHz, CDCl_3_) δH 7.84 (1 H, s, H-1), 6.99 (1 H, s, H-4), 6.27 (1 H, d, *J* = 15.2 Hz, H-9), 6.94 (1 H, d, *J* = 15.6 Hz, H-10), 5.70 (1 H, d, *J* = 9.7 Hz, H-12), 2.47 (1 H, m, H-13), 1.43 (1 H, m, H-14a), 1.33 (1 H, m, H-14b), 0.87 (3 H, t, *J* = 7.4 Hz, H-15), 1.00 (3 H, d, *J* = 6.9 Hz, H-16), 1.84 (3 H, s, H-17), 1.53 (3 H, s, H-18), 2.14 (3 H, s, H-20), 4.55 (2 H, d, *J* = 2.3 Hz, H-1′), 2.63 (1 H, t, *J* = 2.3 Hz, H-3′); ^13^C NMR (500 MHz, CDCl_3_) δ_C_ 140.7 (CH, C-1), 144.2 (C, C-3), 111.7 (CH, C-4), 147.7 (C, C-4a), 103.5 (C, C-5), 184.9 (C, C-6), 85.0 (C, C-7), 193.7 (C, C-8), 115.0 (C, C-8a), 114.6 (CH, C-9), 145.6 (CH, C-10), 132.0 (C, C-11), 148.6 (CH, C-12), 35.3 (CH, C-13), 30.2 (CH_2_, C-14), 12.2 (CH3, C-15), 20.4 (CH_3_, C-16), 12.7 (CH_3_, C-17), 23.3 (CH_3_, C-18), 170.4 (C, C-19), 20.5 (CH_3_, C-20), 43.8 (CH_2_, C-1′), 75.2 (C, C-2′), 77.7 (CH, C-3′); ESI-HRMS *m*/*z* [M + H]^+^ 428.1638 (calcd for C_24_H_26_ClNO_4_H^+^, 428.1623, err. −3.5 ppm).

### 2.5. Semisynthesis of Compound **24**, **39** and **50**

Compounds **24′**, **39′** and **50′** were synthetized following the protocol from Liu et al. [22]. Briefly, a mixture of sclerotiorin (**1**) (2.6 mM) and reactive amine (5.2 mM) was stirred in ethanol aqueous solution (50% H_2_O/50% Ethanol) in the presence of phosphate (0.2 M, pH7) for 120 h at 30 °C, until sclerotiorin (**1**) was completely consumed (monitored by thin layer chromatography, developing solvent: chloroform/methanol/H_2_O = 56/10/0.9). The mixture reaction was evaporated to dryness under reduced pressure. Compounds **24′** and **39′** were purified by liquid–liquid extraction (chloroform/H_2_O) and compound **50′** by preparative thin layer chromatography (developing solvent: chloroform/methanol/H_2_O = 56/10/0.9). 

Isochromophilone leucine (**24′**): Red amorphous oil; [α]^20^_D_ 240 (*c* 0.1 g.L^−1^, MeOH), UV (MeOH) λmax (ε), 382 nm (5480 L.mol^−1^.cm^−1^), 471 nm (1450, L.mol^−1^.cm^−1^); ^1^H NMR (500 MHz, CD_3_CN) δ_H_ 8.03 (1 H, s, H-1), 6.85 (1 H, s, H-4), 6.47 (1 H, d, *J* = 15.6 Hz, H-9), 6.90 (1 H, d, *J* = 15.3 Hz, H-10), 5.65 (1 H, d, *J* = 9.1 Hz, H-12), 2.49 (1 H, m, H-13), 1.41 (1 H, m, H-14a), 1.29 (1 H, m, H-14b), 0.84 (3 H, t, *J* = 6.2 Hz, H-15), 0.99 (3 H, d, *J* = 6.7 Hz, H-16), 1.83 (3 H, s, H-17), 1.43 (3 H, s, H-18), 2.20 (3 H, s, H-20), 4.69 (1 H, m, H-1′), 2.06 (2 H, m, H-3′), 1.27 (1 H, m, H-4′), 0.84 (3 H, m, H-5′), 0.84 (3 H, m, H-6′); ^13^C NMR (700 MHz, CD_3_CN) δ_C_ 141.4 (CH, C-1), 151.8 (C, C-3), 112.0 (CH, C-4), 145.1 (CH, C-4a), 100.7 (C, C-5), 184.2 (C, C-6), 86.4 (C, C-7), 194.8 (C, C-8), 116.1 (C, C-8a), 115.1 (CH, C-9), 146.1 (CH, C-10), 133.6 (C, C-11), 147.2 (CH, C-12), 35.5 (CH, C-13), 30.9 (CH_2,_ C-14), 12.3 (CH_3_, C-15), 20.6 (CH_3_, C-16), 13.0, (CH_3_, C-17), 24.0 (CH_3_, C-18), 170.7 (C, C-19), 20.6 (CH_3_, C-20), 42.7 (CH, C-1′), 173.9 (C, C-2′), 30.4 (CH_2_, C-3′), 25.9 (CH, C-4′), 22.1 (CH_3_, C-5′), 23.3 (CH_3_, C-6′); ESI-HRMS *m*/*z* [M + H]^+^ 504.2149 (calcd for C_27_H_34_ClNO_6_H^+^, 504.2147, err. −0.39 ppm).

Isochromophilone phenylethylamine (**39′**): Red amorphous oil; [α]^20^_D_ 140 (*c* 0.1 g.L^−1^, MeOH), UV (MeOH) λmax (ε), 368 nm (12,550 L.mol^−1^.cm^−1^), 466 nm (3350, L.mol^−1^.cm^−1^), ^1^H NMR (500 MHz, CDCl_3_) δ_H_ 7.58 (1 H, s, H-1), 7.00 (1 H, s, H-4), 6.00 (1 H, d, *J* = 15.3 Hz, H-9), 6.94 (1 H, d, *J* = 15.4 Hz, H-10), 5.71 (1 H, d, *J* = 10.0 Hz, H-12), 2.52 (1 H, m, H-13), 1.47 (1 H, m, H-14a), 1.37 (1 H, m, H-14b), 0.91 (3 H, t, *J* = 7.4 Hz, H-15), 1.05 (3 H, d, J = 6.9 Hz, H-16), 1.84 (3 H, s, H-17), 1.55 (3 H, s, H-18), 2.19 (3 H, s, H-20), 4.17 (1 H, m, H-1a’), 4.01 (1 H, m, H-1b’), 3.07 (2 H, m, H-2′), 7.34 (1 H, d, *J* = 7.2 Hz, H-4′), 7.13 (1 H, d, *J* = 7.4 Hz, H-5′), 7.30 (1 H, d, *J* = 7.2 Hz, H-6′), 7.13 (1 H, d, *J* = 7.4 Hz, H-7′), 7.34 (1 H, d, *J* = 7.2 Hz, H-8′); ^13^C NMR (500 MHz, CDCl_3_) δ_C_ 141.0 (CH, C-1), 144.5 (C, C-3), 111.8 (CH, C-4), 147.9 (C, C-4a), 102.7 (C, -5), 184.6 (C, C-6), 85.0 (C, C-7), 194.0 (C, C-8), 114.8 (C, C-8a), 114.8 (CH, C-9), 145.1 (CH, C-10), 131.7 (C, C-11), 148.2 (CH, C-12), 35.2 (CH, C-13), 30.2 (CH_2_, C-14), 12.2 (CH3, C-15), 20.5 (CH_3_, C-16), 12.8 (CH_3_, C-17), 23.4 (CH_3_, C-18), 170.2 (C, C-19), 20.5 (CH_3_, C-20), 55.6 (CH_2_, C-1′), 36.8 (CH_2_, C-2′), 136.0 (C, C-3′), 128.8, (CH, C-4′), 129.5 (CH, C-5′), 127.9 (CH, C-6′), 129.5 (CH, C-7′), 128.8 (CH, C-8′) ESI-HRMS *m*/*z* [M + H]^+^ 494.2093 (calcd for C_29_H_32_ClNO_4_H^+^, 494.2093, err. 0.0 ppm).

Isochromophilone tryptophan (**50′**): Red amorphous oil; [α]^20^_D_ −30 (*c* 0.1 g.L^−1^, MeOH), UV (MeOH) λmax (ε), 382 nm (6450 L.mol^−1^.cm^−1^), 468 nm (2400, L.mol^−1^.cm^−1^); ^1^H NMR (500 MHz, CD_3_CN) δ_H_ 8.4 (1 H, s, H-1), 6.85 (1 H, s, H-4), 6.46 (1 H, d, *J* = 15.4 Hz, H-9), 6.75 (1 H, d, *J* = 15.2 Hz, H-10), 5.45 (1 H, d, *J* = 8.2 Hz, H-12), 2.40 (1 H, m, H-13), 1.40 (1 H, m, H-14a), 1.30 (1 H, m, H-14b), 0.87 (3 H, t, *J* = 7.3 Hz, H-15), 0.95 (3 H, d, *J* = 6.3 Hz, H-16), 1.50 (3 H, s, H-17), 1.41 (3 H, s, H-18), 2.14 (3 H, s, H-20), 3.91 (1 H, m, H-1′), 3.30 (2 H, m, H-3′), 5.15 (1 H, m, H-5′), 7.31 (1 H, d, *J* = 7.8 Hz, H-8′), 7.01 (1 H, t, *J* = 7.7 Hz, H-9′), 7.08 (1 H, t, *J* = 7.1 Hz, H-10′), 7.63 (1 H, d, *J* = 7.8 Hz, H-11′); ^13^C NMR (500 MHz, MeOD) δ_C_ 138.8 (CH, C-1), 147.7 (C, C-3), 124.9 (CH, C-4), 148.4 (CH, C-4a), 102.1 (C, C-5), 185.4 (C, C-6), 86.5 (C, C-7), 195.4 (C, C-8), 117.5 (C, C-8a), 112.1 (CH, C-9), 146.0 (CH, C-10), 133.8 (C, C-11), 147.7 (CH, C-12), 36.2 (CH, C-13), 31.3 (CH_2,_ C-14), 12.5 (CH_3_, C-15), 20.7 (CH_3_, C-16), 12.5, (CH_3_, C-17), 24.0 (CH_3_, C-18), 171.7 (C, C-19), 20.4 (CH_3_, C-20), 56.6 (CH, C-1′), 174.1 (C, C-2′), 28.6 (CH_2_, C-3′), 109.4 (C, C-4′), 68.9 (CH, C-5′), 112.6 (CH, C-8′), 120.6 (CH, C-9′), 122.9 (CH, C-10′), 119.2 (CH, C-11′), 128.3 (C, C-12′); ESI-HRMS *m*/*z* [M + H]^+^ 577.2092 (calcd for C_32_H_33_ClN_2_O_6_H^+^, 577.2100, err. 1.39 ppm).

### 2.6. UV-Visible Spectrophotometry and Optical Density (OD_nm_) Calculation of Crude Extract 

Dried crude extracts were resuspended at 1 mg.mL^−1^ in analytical grade methanol. UV spectra for crude extract were recorded at 20 °C in analytical grade methanol using a PerkinElmer Lambda 5 spectrophotometer. Acquisition was made from 800 nm to 200 nm with scan interval of 1 nm and scan rate of 600 nm.min^−1^. Light source changeover from visible to UV was made at 350 nm. The OD yield at 400, 460 and 500 nm was obtained using the following formula:

OD_nm_ = (Absorbance_nm)_ × (Dilution factor) × (Dried Crude extract mass)/(Petri dish mass), and is given in arbitrary unit_nm_ per gram of solid medium (AU_nm_.g^−1^ of solid medium).

### 2.7. LC-MS/MS Analysis

Crude extracts of *Penicillium sclerotiorum* SNB-CN111 were prepared at 1 mg.mL^−1^ in methanol and filtered on 0.45 µm PTFE membranes. LC-MS/MS experiments were performed with a 1260 Prime HPLC (Agilent Technologies, Waldbronn, Germany) coupled with an Agilent 6540 Q-ToF (Agilent Technologies, Waldbronn, Germany) tandem mass spectrometer. LC separation was achieved with an Accucore RP-MS column (100 × 2.1 mm, 2.6 μm, Thermo Scientific, Les Ulis, France) with a mobile phase consisting of H_2_O/formic acid (99.9/0.1) (A)—acetonitrile/formic acid (99.9/0.1) (B). The column oven was set at 45 °C. Compounds were eluted at a flow rate of 0.4 mL.min^−1^ with a gradient from 5% B to 100% B in 20 min and then 100% B for 3 min. Injection volume was fixed at 5 μL for all analyses. For electrospray ionization source, mass spectra were recorded in positive ion mode with the following parameters: gas temperature 325 °C, drying gas flow rate 10 L.min^−1^, nebulizer pressure 30 psi, sheath gas temperature 350 °C, sheath gas flow rate 10 L.min^−1^, capillary voltage 3500 V, nozzle voltage 500 V, fragmentor voltage 130 V, skimmer voltage 45 V, and Octopole 1 RF Voltage 750 V. Internal calibration was achieved with two calibrants, purine and hexakis (1h,1h,3h-tetrafluoropropoxy) phosphazene (*m*/*z* 121.0509 and *m*/*z* 922.0098), providing a high mass accuracy better than 5 ppm. The data-dependent MS/MS events were acquired for the five most intense ions detected by full-scan MS, from *m*/*z* 200–1000, above an absolute threshold of 1000 counts. Selected precursor ions were fragmented at a fixed collision energy of 30 eV and with an isolation window of 1.3 amu. The mass range of the precursor and fragment ions was set as *m*/*z* 200–1000.

Isolated compounds from *Penicillium sclerotiorum* SNB-CN111 fractions were prepared at 0.1 mg.mL^−1^ in methanol and filtered on 0.45 µm PTFE membrane for exact mass measurements. 

### 2.8. Data Processing and Analysis

The data files were converted from the .d standard data format (Agilent Technologies) to .mzXML format using the MSConvert software, part of the ProteoWizard package 3.0 [30]. All .mzxml were processed using MZmine2v51 as previously described [31]. The mass detection was realized with MS1 noise level at 1000 and MS/MS noise level at 0. The ADAP chromatogram builder was employed with a minimum group size of scans of 3, a group intensity threshold of 1000, a minimum highest intensity of 1000, and *m*/*z* tolerance of 0.008 (or 20 ppm). Deconvolution was performed with the ADAP wavelets algorithm according to the following settings: S/N threshold = 10, minimum features height = 1000, coefficient/area threshold = 10, peak duration range 0.01–1.5 min, *t_R_* wavelet range 0.00–0.05 min. MS/MS scans were paired using an *m*/*z* tolerance range of 0.05 Da and *t_R_* tolerance range of 0.5 min. Isotopologues were grouped using the isotopic peak grouper algorithm with an *m*/*z* tolerance of 0.008 (or 20 ppm) and a *t_R_* tolerance of 0.2 min. Peaks were filtered using Feature list row filter, keeping only peaks with MS/MS scans (GNPS). Peak alignment was performed using the join aligner with an *m*/*z* tolerance of 0.008 (or 20 ppm), a weight for *m*/*z* at 20, a *t_R_* tolerance of 0.2 min, and weight for *t_R_* at 50. The MGF file and the metadata were generated using the export/submit to GNPS option. 

Molecular networks were calculated and visualized using MetGem 1.3 software [26,27,28], MS/MS spectra were window-filtered by choosing only the top 6 peaks in the ±50 Da window throughout the spectrum. The data were filtered by removing all peaks in the ±17 Da range around the precursor *m*/*z*. The *m*/*z* tolerance windows used to find the matching peaks were set to 0.02 Da, and cosine scores were kept in consideration for spectra sharing 2 matching peaks at least. The number of iterations, perplexity, learning-rate, and early exaggeration parameters were set to 5000, 25, 200, and 12 for t-SNE view. 

Figures were generated using R and related packages (ggplot2, Rcolorbrewer, VennDiagram and gridextra), MetGem export function, and ChemDraw Professional 16.0 (PerkinElmer). NMR spectra were processed and analyzed using TopSpin 3.6.2 (Bruker, Rheinstetten, Germany).

## 3. Results and Discussion

### 3.1. Increasing Azaphilone Red Pigments Diversity

To investigate in vivo incorporation of amines into azaphilones produced by *P. sclerotiorum* SNB-CN111, 3 solid media with increasing nitrogen complexity were prepared. It included the Czapek medium (CZK), a minimal medium with only 2 g.L^−1^ of sodium nitrate as inorganic nitrogen source, potato dextrose (PD) a complex medium, whose organic nitrogen source comes from 4 g.L^−1^ of potato infusion, and finally, a medium with high organic nitrogen complexity, potato dextrose medium supplemented with 20 g.L^−1^ of yeast extract (YE). After 15 days of cultivation, ethyl acetate extracts for each condition were performed and subjected to reverse-phase liquid chromatography hyphenated to positive electrospray ionization tandem mass spectrometry (RPLC-ESI(+)-MS/MS) to evaluate the complexity of the metabolomic profiles (Figure 1a). MS/MS data were then extracted to generate molecular networks by MetGem software [26,27,28]. In a molecular network, specialized metabolites of the same family cluster according to their fragmentation similarity, facilitating the annotation of metabolites in complex mixtures. The advantage of t-SNE molecular networks over regular molecular network is that datasets are represented as a whole, preserving the structural relationship between distinct clusters. This methodology has already been successfully applied to extracts of *Penicillium sclerotiorum* SNB-CN111 on solid PD conditions to decipher the chemical diversity of produced azaphilones [24,25]. Cultivation on solid CZK medium, PD medium, and solid YE medium generated 255, 620 and 1069 distinct metabolic features, respectively, indicating that the chemical diversity is increasing with the complexity and concentration of the organic nitrogen sources (Figure 1b). Increasing the complexity of the organic nitrogen source also affected the production of unique metabolites. In fact, cultivation on YE led to the production of 716 unique features, accounting for 67% of YE produced metabolites. On the other hand, 29% and 35% of compounds produced by cultivation on CZK and PD were specific to these conditions, respectively.

To explore the t-SNE molecular network, a MS/MS database built from a previous work [25] was used leading to the annotation of 39 azaphilones distributed in five clusters (Figure 1c). Yellow and orange azaphilones with pyran ring were annotated in clusters A and B. Cluster A displays the analogous metabolites of sclerotiorin (**1**) and cluster B the ochrephilone ones (**2**) (Appendix A, Appendix A). Thus, structural diversity of cluster A and B is not significantly modified by the nature of the nitrogen source as most abundant ions are equally produced from cultivation on CZK, PD or YE media (Figure 1). Cluster C is constituted of red azaphilone pigments sclerotioramine (**3**) and its analogs resulting from the conversion of the pyran oxygen with ammonia to form red vinylogous γ-pyridone (Appendix A and Appendix A) [14]. Organic nitrogen sources offered by PD and YE media are thus necessary to induce formation of sclerotioramine (**3**) analogs. Finally, clusters D and E are related to red and purple analogs of nitrogen-incorporated sclerotiorin (**1**) (cluster D) or ochrephilone (**2**) (cluster E) (Appendix A, Appendix A). The complexity of nitrogen sources directly influences the diversity of red azaphilones pigments. Indeed, out of the 174 azaphilones constituting cluster D, only 1 is produced from CZK medium, 28 are produced from PD and 162 azaphilones are produced from YE medium.

As the organic nitrogen in YE results from protein digestion and amino acid metabolism, MS and MS/MS data of protonated species corresponding to putative analogs of sclerotiorin (**1**) with integrated amino acids or their metabolic derivatives were scrutinized. The molecular formula of the functionalized nitrogen chain was calculated and queried in the KEGG compound database [32] in order to annotate metabolites (Appendix A). Cluster D was subdivided into three clusters (D-I, D-II and D-III) to facilitate the annotation in the t-SNE molecular network (Appendix A). Fourteen azaphilones are bearing an amino acid as a nitrogen side chain, some over can be related to amino acid derivatives such as ethanolamine or tyramine (**6–50**) (Figure 2). The same methodology was performed for cluster E allowing the annotation of 13 purple azaphilone pigments (**5**, **51–62**) (Appendix A and Appendix A). Surprisingly, red and purple azaphilones with ethanolamine moiety (**4** and **52**) are produced in solid CZK medium. This may correlate with the fact that ethanolamine is a precursor of phospholipids which are part of biological membranes [33]. Then, the structural annotations of cluster D were confirmed by hemi-synthesis of sclerotiorin derivatives for leucine **24′**, phenylethylamine **39′** and tryptophan **50′** (Appendix A). Identical retention times and MS/MS profiles, with cosine scores of 0.99, 1.00 and 0.98, respectively, confirmed the first annotation of the t-SNE molecular network (Appendix A).

By increasing the diversity of organic nitrogen sources in the solid culture media, the diversity of red azaphilone pigments drastically increased, confirming the great versatility of the sclerotiorin azaphilone scaffold to incorporate functionalized nitrogen [34]. Additionally, the absorbance at 500 nm (AU_500_.g^−1^of solid medium) of the crude extract from YE cultivation increased by a factor of 10 to reach 59.6 AU_500_.g^−1^ of solid medium compare to crude extract from PD cultivation. This value can be considered as high compared to the literature (Appendix A) [23]. 

As the CZK medium is leading to poor production of nitrogen azaphilones, the addition of one single organic nitrogen source into this culture medium should unlock the production of specific azaphilone red pigments, as it was previously demonstrated for submerged cultivation [21,22]. 

### 3.2. Selective Overproduction of Red Azaphilone Pigments

Solid CZK medium was prepared with a single organic nitrogen source, i.e., ammonium acetate and ethanolamine, to overproduce metabolites **3** or **4**, the main red azaphilone pigments produced by *P. sclerotiorum* SNB-CN111 in solid PD and YE media. Unfortunately, no production operates when performing this process (Appendix A). RPLC-ESI(+)-MS analyses further confirmed that molecule **3** was not produced at all, whereas molecule **4** is slightly overproduced by a factor of 2 to 4 for both nitrogen sources. Moreover, the AU_500_.g^−1^ of solid medium of the crude extract remains unchanged (Appendix A and Appendix A). Therefore, the protocol was adapted by growing *P. sclerotiorum* SNB-CN111 on a porous cellophane membrane for 7 days with a solid CZK medium, then shifting the membrane to a solid CZK medium complemented with a specific organic nitrogen source for 7 additional days (Figure 3a).

Using this two-step solid-state cultivation, the overall azaphilone production with addition of either ammonium acetate or ethanolamine was successfully increased as the absorbance of the crude extracts at AU_400_.g^−1^, AU_460_.g^−1^ and AU_500_.g^−1^ of the medium matched those obtained from solid PD cultivation (Appendix A). RPLC-ESI(+)-MS analyses of the ethyl acetate extracts confirmed the specific overproduction of molecules **3** or **4** depending on the concentration of organic nitrogen sources (Figure 3b). The versatility of this method was further investigated by integrating propargylamine, a xenobiotic into the azaphilone scaffold.

### 3.3. Selective Overproduction of Red Azaphilone Pigments

The molecule **63** with propargylamine moiety has already been hemi-synthesized by addition of functionalized nitrogen to **1** within trimethylamine and potassium carbonate solution. Compound **63** possesses an anti-fouling effective concentration of 0.94 µg.mL^−1^ [34]. In addition, alkyne groups allow click chemistry extending the range of applications of red azaphilone pigments [35,36]. Using the two-step solid cultivation on membrane, molecule **63** is overproduced in a concentration dependent manner (Figure 4). In addition, OD_500_ values of the crude extracts from solid CZK supplemented with propargylamine at a concentration of 0.5 and 2 g.L^−1^ matched the value obtained with solid YE medium with 68 and 50 AU_500_.g^−1^ of solid medium, respectively (Appendix A).

The CZK medium allows generating extracts with a considerably lower azaphilone diversity, especially for red azaphilones pigments that are more polar than their yellow and orange analogs (Appendix A). As a result, the isolation of specifically produced red azaphilone pigments is facilitated. Molecule **63** was isolated from 120 mg of pooled ethyl acetate crude extracts corresponding to 4 Petri dishes of 80 cm^2^. A first flash chromatography experiment allowed us to recover 30 mg of a fraction enriched in molecule **63** (Appendix A). Nineteen milligrams of molecule **63** were then isolated by reverse phase (C18) preparative HPLC, leading to a yield of about 16% from the crude extract. The structure of compound **63** was confirmed by ^1^H, ^13^C, COSY, HSQC and HMBC NMR experiments and compared with the literature and previously isolated red azaphilones **3** and **4** (Appendix A) [25,37]. 

## 4. Conclusions

The production of azaphilones from *P. sclerotiorum* SNB-CN111 was explored using different culture media with various types and concentrations of nitrogen sources. Annotations using t-SNE molecular networks allowed the investigation of the chemical diversity in the different conditions. These annotations were subsequently validated by the hemi-synthesis of three analogues of sclerotiorin that incorporated biogenic amines and amino acids. Moreover, the addition of organic nitrogen in solid medium provided crude extracts with high visible light absorbance at 500 nm. A two-step cultivation process on solid medium was then developed. By growing the strain on cellophane with a single source of inorganic nitrogen, only yellow and orange azaphilones with a pyran core were overproduced. The transfer of the microorganism to a second medium enriched in organic amine allowed overproduction of the desired functionalized sclerotiorin. The proof of concept was then complemented by overproducing a red azaphilone derived from sclerotiorin, which incorporated propargylamine, with high yield (16%). 

By extending the two-step cultivation methods to solid media, more fungal strains can be tested for their ability to overproduce azaphilone red pigments. Being able to choose the sources of nitrogen incorporated in azaphilones will enable to modulate their colors, reduce the amount of nitrogen needed in the medium or the cost of the fermentative process, contributing to the attractiveness of azaphilones as food grade pigments.

## Figures and Tables

**Figure 1 jof-09-00156-f001:**
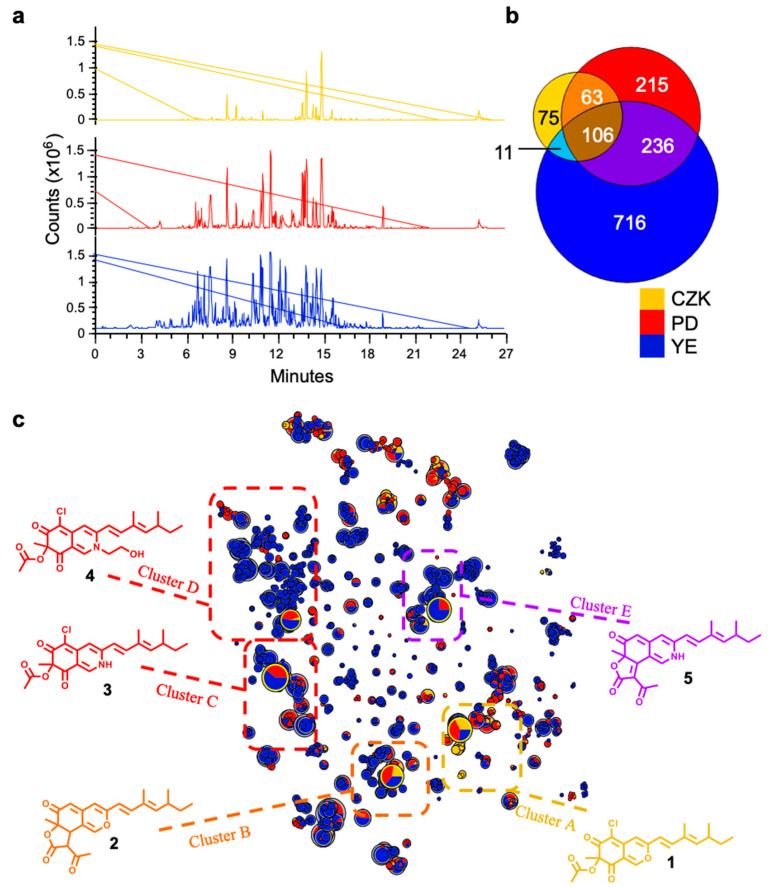
Azaphilone diversity from *P. sclerotiorum* SNB-CN111 crude extracts induced by an increase in organic nitrogen complexity in the solid medium from CZK medium (yellow, no organic nitrogen), PD medium (red, medium organic nitrogen complexity) to YE (blue, high organic complexity). (**a**) Metabolic profile from RPLC-ESI(+)-MS/MS analysis of crude extracts (TIC values). (**b**) Shared and unique features between metabolic profile of *P. sclerotiorum* SNB-CN111 cultivation on solid CZK, PD and YE medium. (**c**) t-SNE molecular network representation constructed on MS/MS features homology with the size of node related to their intensity and color of the node to cultivation medium with dereplicated features and structure related to their pigment color. Standards **1**, **2**, **3**, **4** and **5** previously isolated [25] are circled in gold.

**Figure 2 jof-09-00156-f002:**
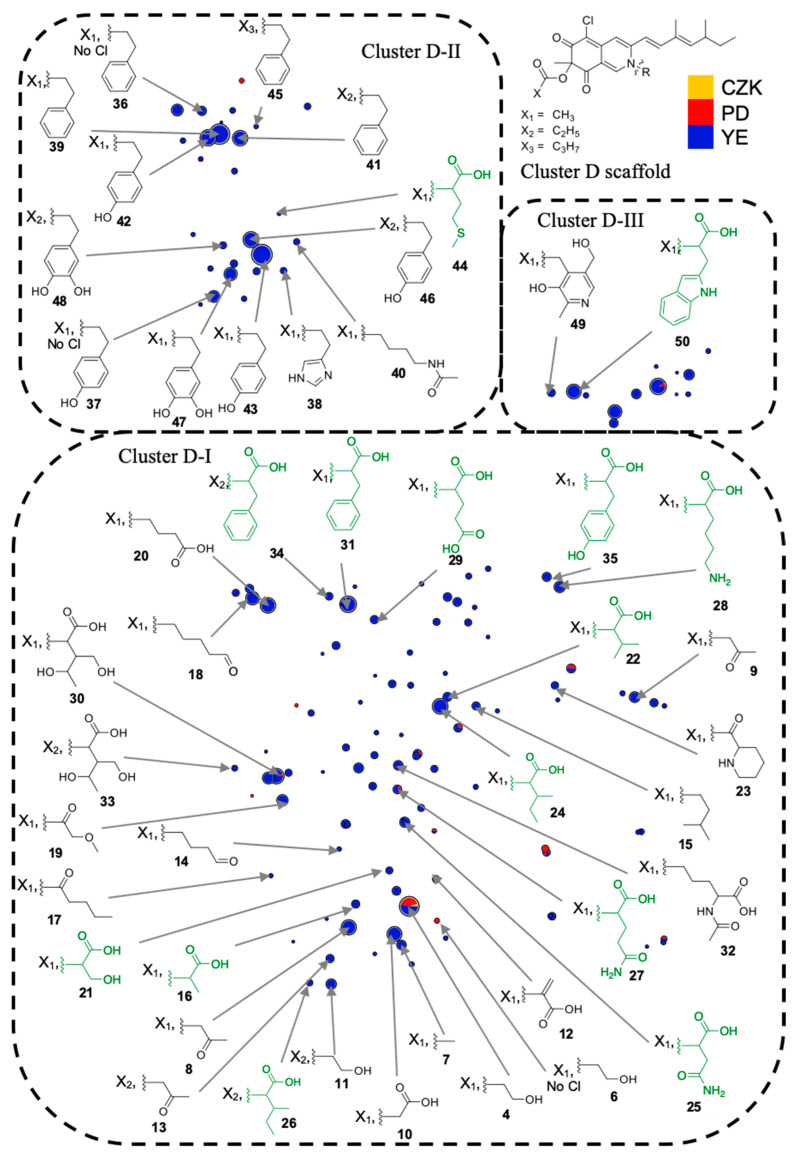
Annotation of the clusters D-I-, D-II and D-III with functionalized nitrogen chain structures integrated to the azaphilone scaffold, proposed from molecular network data and KEGG compound database. Acylation (X_1_), proprionylation (X_2_) and butylation (X_3_) were identified using characteristic neutral losses [25]. “No Cl” means no chlorination moiety on the azaphilone scaffold; R amino acid groups are indicated in green.

**Figure 3 jof-09-00156-f003:**
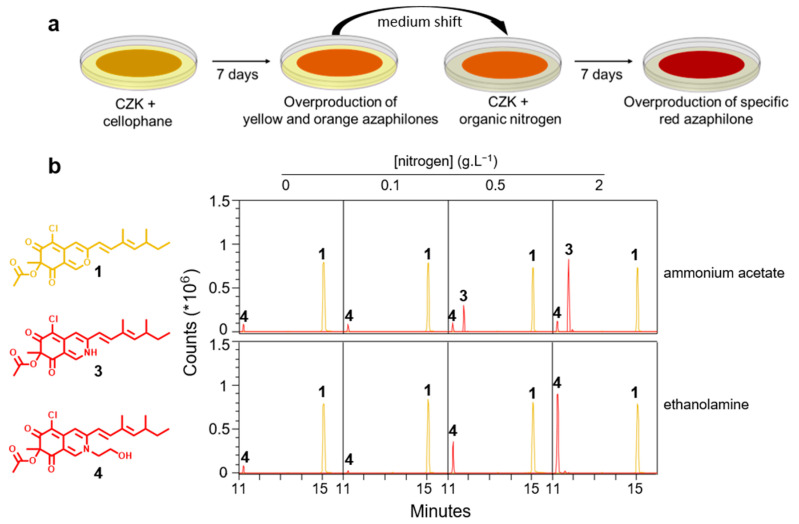
Specific incorporation of functionalized nitrogen into azaphilones. (**a**) Two-step solid-state cultivation on porous Cellophane membrane to achieve specific red azaphilone pigments overproduction by *P. sclerotiorum* SNB-CN111 and (**b**) extracted ion chromatograms for molecules **1** ([M + H]^+^, *m*/*z* 391.1307), **3** ([M + H]^+^, *m*/*z* 390.1467) and **4** ([M + H]^+^, *m*/*z* 434.1729) as a function of the concentration of ammonium acetate or ethanolamine added to CZK medium.

**Figure 4 jof-09-00156-f004:**
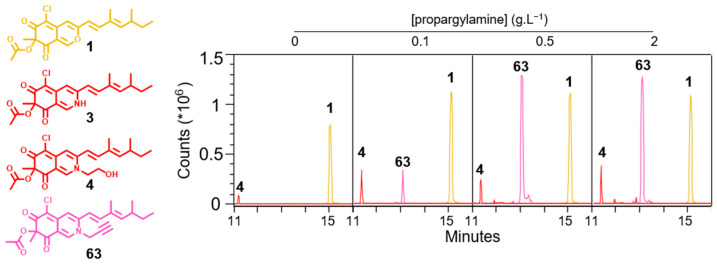
Overproduction of compound **63** by *P. sclerotiorum* SNB-CN111 with two-step solid state cultivation on porous cellophane membrane. Extracted ion chromatogram of compounds **1** ([M + H]^+^, *m*/*z* 391.1307), **3** ([M + H]^+^, *m*/*z* 390.1467), **4** ([M + H]^+^, *m*/*z* 434.1729) and **63** ([M + H]^+^, *m*/*z* 428.1638) in function of propargylamine concentration added to solid CZK medium.

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
