# Peer review of "Nitrogen Enriched Solid-State Cultivation for the Overproduction of Azaphilone Red Pigments by Penicillium sclerotiorum SNB-CN111"

_jof, 2023, doi:10.3390/jof9020156_

Round 1

Reviewer 1 Report

The differences of reference 25 and the manuscript should be described. 

Author Response

Reference 25 is dedicated to the dereplication and annotation of azaphilones from SNB-CN111 cultures in classical PDA medium.

In the proposed manuscript, azaphilone production was oriented using OSMAC strategy so that the SNB-CN111 strain produced mainly red pigments. These pigments are due to the incorporation of nitrogen during the biosynthesis of these secondary metabolites. This stimulation of the azaphilone biosynthetic pathway with nitrogen groups was performed using an OSMAC method including different nitrogen sources in the culture media. As described in line 55-60 “For that purpose, we developed of an unique two-step nitrogen enriched solid-state cultivation method able to overproduce red azaphilone pigments and their chemical diversity related to the nitrogen … offering a new pipeline for large-scale red azaphilone production.”

Moreover, the submitted manuscript proposed a new protocol to overproduce nitrogen-containing azaphilone using a new two-steps strategy.

Thus reference 25 and the submitted paper are both related to the same strains but with very different scientific questions and strategies.

Reviewer 2 Report

The manuscript reported a new two-step solid-state cultivation process to produce specific red azaphilones pigments and their chemical diversity was explored based on liquid chromatography coupled to tandem mass spectrometry (LC-MS/MS) and molecular network. This two-step procedure first implies a cellophane membrane allowing accumulating yellow and orange azaphilones from a Penicillium sclerotiorum SNB-CN111 strain, and second involves the incorporation of the desired functionalized nitrogen by shifting the culture medium. However, many issues should be addressed before publication.

Minor issues:

1. Page 6, Figure 1a, is the Y-axis of Figure 1a absorbance? If so, please mark wavelength. Please check other figures as well.

2. Page 6, line 267, “706 unique features, accounting for 69% of YE produced metabolites.”

Is it 706 or 716? Please be consistent with Figure 1b. Please confirm that the other data in the full text is correct.

3. Page 7, line 293, “or ochrephilone (3) (cluster E)”

  Whether it should be “or ochrephilone (2) (cluster E)”? Please check it.

4. Page 7, line 303, “Fifteen azaphilones are bearing an amino acid as a nitrogen side chain, 28 are related to amino acid derivatives such as ethanolamine or tyramine, and 2 to intermediates in lipid synthesis (6-50) (Figure 2).”

I think it’s better to change “Fifteen” to “15”.

In addition, it’s better to draw the three different types of azaphilones in three different colors.

5. In the process of data analysis, whether other new azaphilones are found to be produced? If so, it can be shown in a list or a figure.

Author Response

We thank you for evaluating our article, we have improved it with your comments; see below.

Minor issues:

  1. Page 6, Figure 1a, is the Y-axis of Figure 1a absorbance? If so, please mark wavelength. Please check other figures as well.

On the Y-axis of all the LC-MS figures, the TICs (Total Ion Current chromatogram) are shown. The total ion current or counts (TIC) chromatogram represents the summed intensity across the entire range of masses being detected at every point in the analysis.

  1. Page 6, line 267, “706 unique features, accounting for 69% of YE produced metabolites.”

Is it 706 or 716? Please be consistent with Figure 1b. Please confirm that the other data in the full text is correct.

Some typos appeared in the paragraph, we have corrected them.

  1. Page 7, line 293, “or ochrephilone (3) (cluster E)”

Whether it should be “or ochrephilone (2) (cluster E)”? Please check it.

Thank you for your observation, in fact ochrephilone corresponds to compound 2 and not 3.

  1. Page 7, line 303, “Fifteen azaphilones are bearing an amino acid as a nitrogen side chain, 28 are related to amino acid derivatives such as ethanolamine or tyramine, and 2 to intermediates in lipid synthesis (6-50) (Figure 2).”

I think it’s better to change “Fifteen” to “15”.

In addition, it’s better to draw the three different types of azaphilones in three different colors.

The paragraph and Figure 2 related to the latter have been modified to clarify the latter.

  1. In the process of data analysis, whether other new azaphilones are found to be produced? If so, it can be shown in a list or a figure.

All the identified and annotated molecules are described in the Supplementary data Figure S1 to S7 and the new compounds correspond to those not named in Tables 1 to 8.

Reviewer 3 Report

The study Nitrogen enriched solid-state cultivation for the overproduction of azaphilone red pigments by Penicillium sclerotiorum SNB-CN111 Is well designed, written and discussed. The authors developed unique two-step nitrogen enriched solid-state cultivation method able to overproduce red azaphilone pigments and their chemical diversity related to the nitrogen sources was characterized using liquid chromatography coupled to LC-MS/MS. The data is very important for the future outlook of Azaplilone production.

Author Response

We would like to thank you for reviewing our article.

Round 2

Reviewer 1 Report

After revision, it can be accepted.